# Preterm Birth Impedes Structural and Functional Development of Cerebellar Purkinje Cells in the Developing Baboon Cerebellum

**DOI:** 10.3390/brainsci10120897

**Published:** 2020-11-24

**Authors:** Tara Barron, Jun Hee Kim

**Affiliations:** The Department of Cellular and Integrative Physiology, University of Texas Health Science Center, San Antonio, TX 78229, USA; tbarron@stanford.edu

**Keywords:** non-human primate, baboon, cerebellum, fetal development, preterm birth, Purkinje cell, electrophysiology, NICU

## Abstract

Human cerebellar development occurs late in gestation and is hindered by preterm birth. The fetal development of Purkinje cells, the primary output cells of the cerebellar cortex, is crucial for the structure and function of the cerebellum. However, morphological and electrophysiological features in Purkinje cells at different gestational ages, and the effects of neonatal intensive care unit (NICU) experience on cerebellar development are unexplored. Utilizing the non-human primate baboon cerebellum, we investigated Purkinje cell development during the last trimester of pregnancy and the effect of NICU experience following premature birth on developmental features of Purkinje cells. Immunostaining and whole-cell patch clamp recordings of Purkinje cells in the baboon cerebellum at different gestational ages revealed that molecular layer width, driven by Purkinje dendrite extension, drastically increased and refinement of action potential waveform properties occurred throughout the last trimester of pregnancy. Preterm birth followed by NICU experience for 2 weeks impeded development of Purkinje cells, including action potential waveform properties, synaptic input, and dendrite extension compared with age-matched controls. In addition, these alterations impact Purkinje cell output, reducing the spontaneous firing frequency in deep cerebellar nucleus (DCN) neurons. Taken together, the primate cerebellum undergoes developmental refinements during late gestation, and NICU experience following extreme preterm birth influences morphological and physiological features in the cerebellum that can lead to functional deficits.

## 1. Introduction

Preterm birth is prevalent even in developed countries, as nearly 10% of infants were born preterm in the US in 2017 [1]. After pre-term birth, infants continue their development in the highly stimulatory environment of the neonatal intensive care unit (NICU), rather than protected in the womb. Although survival of preterm infants has increased due to improved treatment in the NICU, preterm birth still results in deficits in neurodevelopment, leading to long-term disabilities. The cerebellum is a brain region that is severely impacted by preterm birth, likely due to its development late in gestation, when preterm birth occurs [2]. Individuals born prematurely have been shown a reduction in cerebellar size associated with hemorrhage [3], infarction [4], or, commonly, underdevelopment [5,6,7]. Reduced cerebellar size caused by preterm birth can lead to cognitive, neuropsychological, and motor function deficits [5,8] and can persist into adulthood [8].

The largest contributor to cerebellar volume is the proliferation and migration of cells from the external granule layer (EGL) [7]. Granule cells migrate from the EGL to internal granule layer (IGL) during development, creating parallel fibers that synapse onto Purkinje cells. This input is critical for Purkinje cell development and survival, as the absence of granule cells results in reduced Purkinje cell differentiation and increased Purkinje cell death [9]. Purkinje neurons are the sole output cells of the cerebellar cortex, sending their axons through the IGL and white matter to synapse onto the deep cerebellar nucleus (DCN). Studies have shown that alterations to Purkinje cell input to DCN cells result in changes in DCN cell activity [10,11,12]. Thus, studying Purkinje cell development throughout late gestation is important for understanding deficits in cerebellar growth in preterm infants. 

Cerebellar development in humans and non-human primates occurs in primarily in utero, and preterm birth results in microstructural alterations in the cerebellum in baboon neonates, such as reduced cerebellar weight and decreased cerebellar granule cell proliferation [13,14]. A histological analysis of postmortem human neonate tissue after preterm birth showed that preterm birth and ex-utero effects altered granule cells and Bergmann glia differentiation in the cerebellum [15]. However, it remains to be elucidated whether gestational age at delivery and NICU experience individually or collectively affect proper development of the physiological properties of Purkinje cells. Single-cell electrophysiological records require tissue in which the cells are sufficiently healthy for neuronal activity to occur, which often cannot be accomplished from human samples. The electrophysiological and morphological development of Purkinje cells has been studied in rodents [16], but the development of the rodent cerebellum occurs primarily postnatally, posing a limitation when attempting to investigate the effects of preterm birth. Using baboon neonates, we performed whole-cell patch clamp recordings to investigate Purkinje cell development at three timepoints throughout the last trimester of pregnancy: extreme preterm (hereafter E-Pre; 70% of gestational age, GA), moderate preterm (hereafter M-Pre; 80% of GA), and full term (hereafter Term). Furthermore, this study demonstrates that 2 weeks development in the NICU after E-Pre birth (hereafter NICU) impedes functional cellular development of Purkinje cells compared to in utero development.

## 2. Materials and Methods

### 2.1. Animals

A total of 22 brains from baboons of either sex were delivered at the Texas Biomedical Research Institute in San Antonio. Animals were used in accordance with approved University of Texas Health Science Center, San Antonio Institutional Animal Care and Use Committee protocols (IACUC Protocol Number: 1567 PC 2). E-Pre baboons were delivered via cesarean section at 125 ± 2 d gestational age (67% of gestation). A total of 5 of these E-Pre baboons were immediately euthanized to constitute the E-Pre group, while 10 were intubated immediately after birth and chronically ventilated for a planned survival of 14 d in the NICU to constitute the NICU group. NICU baboon care has been described in detail [17]. Briefly, central intravenous lines were placed shortly after birth for fluid management and parenteral nutrition, and baboons were treated with corticosteroids, antibiotics, ketamine, valium, vitamin K, and blood transfusions. A total of 4 M-Pre baboons were delivered via cesarean section at 140 ± 2 d gestational age (76% of gestation). A total of 3 term baboons were delivered naturally at approximately 185 d gestational age. 

### 2.2. Slice Preparation

During neonatal baboon necropsy, the cerebellum was removed from the skull and immersed in ice-cold artificial CSF (aCSF) containing the following (in mM): 125 NaCl, 2.5 KCl, 3 MgCl_2_, 0.1 CaCl_2_, 25 glucose, 25 NaHCO_3_, 1.25 NaH_2_PO_4_, 0.4 ascorbic acid, 3 myo-inositol, and 2 Na-pyruvate, pH 7.3–7.4, when bubbled with carbogen (95% O_2_, 5% CO_2_; osmolality of 310–320 mOsm/kg water). For electrophysiology, midsagittal cerebellar slices (300 μm) were prepared using a vibratome (VT1200S, Leica). After cutting, the slices were transferred to an incubation chamber containing normal aCSF bubbled with carbogen and maintained at 35 °C for 30 min and thereafter at room temperature. Normal aCSF was the same as the slicing aCSF, but with 1 mm MgCl_2_ and 2 mm CaCl_2_. For immunohistochemistry, the cerebellum was post-fixed overnight at 4 °C followed by cryoprotection with 30% (*w*/*v*) sucrose in 0.1 M PBS (pH 7.3). Midsagittal cerebellar slices (80 μm) were cut with a microtome cryostat (HM 505E, Microm).

### 2.3. Electrophysiology

Whole-cell patch clamp recordings were performed in normal aCSF at room temperature (22 °C–24 °C). Pipette solution contained the following (in mm): 130 K-gluconate, 20 KCl, 5 Na_2_-phosphocreatine, 10 HEPES, 0.2 EGTA, 4 Mg-ATP, and 0.3 GTP, pH adjusted to 7.3 with KOH. Spontaneous postsynaptic currents (sPSCs) or action potentials (APs) were recorded in normal aCSF using the voltage or current-clamp mode of the EPC-10 (HEKA Electronik). Patch electrodes had resistances of 3–4 mΩ, and the initial uncompensated series resistance (R_s_) was <25 mΩ. APs were elicited by current injection in current-clamp mode. Postsynaptic currents were measured under whole-cell voltage-clamp with a holding potential of −65 mV. Spontaneous postsynaptic currents were analyzed using Mini-analysis software (Synaptosoft Inc., Decatur, GA, USA).

### 2.4. Immunohistochemistry

After slicing with a cryostat, free-floating cerebellar slices (80 μm) were blocked in 4% goat serum and 0.3% Triton X-100 in PBS for 1 h. Rabbit anti-calbindin (CB; 1:200; Cell Signaling Technologies, Cat # 13176) was used as a primary antibody. Staining was reported by incubation with an Alexa dye-conjugated secondary antibody (1:500, Invitrogen) for 2 h at room temperature. After five rinses with PBS, slices were incubated with 4′,6-diamidino-2-phenylindole (DAPI; 1:500, Sigma-Aldrich) to counterstain cell nuclei. Slices were mounted onto Superfrost slides in photobleaching-protective medium (Fluoroshield; Sigma-Aldrich). Stained slices were viewed using a 10× or 20× air objective or a 40× oil-immersion objective on a confocal laser-scanning microscope (LSM-510, Zeiss). Stack images were acquired at a digital size of 1024  ×  1024 pixels with optical section separation (z interval) of 0.5 μm and were later cropped to the relevant part of the field without changing the resolution.

### 2.5. Quantification

Morphological analysis was completed in lobule 8 of midsagittal cerebellar sections. Layer width was measured at 15 locations throughout the layer, each 50 μm apart, from each section. For IGL and EGL cell density quantification, DAPI+ cells were counted within a 50 × 50 μm square in each slice. For Purkinje cell density, CB+ cell bodies were counted along a 200 μm line in the Purkinje cell layer in each slice and averaged per animal. For all cell counts, only cells that were in focus and completely within the square or line were counted. Purkinje cell soma size was quantified by measuring diameter of the cell.

## 3. Results

### 3.1. Morphological Development of the Cerebellum Occurs during Late Gestation and in the NICU

Studying the baboon neonate brain allows direct comparisons with the human brain because the connectivity, size, and functional areas are similar to those in humans. The premature baboon, delivered at 126 days GA or 140 days GA (term = 180 days GA), shares a similar neonatal course with the human preterm birth at 26–28 weeks (70% of GA, E-Pre) or 32 week of GA (80% of GA, M-Pre, Figure 1a), exhibiting common complications relevant to prematurity including incomplete lung development and metabolism [17]. During the last trimester of pregnancy, the cerebellum became larger in size of overall structures with an increased surface area and folial complexity as described in human cerebellum [7,18,19] (Figure 1b).

Granule cells influence structural and functional development and maturation of Purkinje cells [9], which occurs during late gestation in primates. Granule cells migrate from the external granule layer (EGL) to the internal granule layer (IGL) throughout gestation and early infancy, during which time they create parallel fibers that synapse onto Purkinje neurons [7]. We assessed whether the EGL and IGL were altered throughout late gestation in baboon. Labeling cells with DAPI and calbindin clearly revealed the EGL and IGL in the cerebellar cortex throughout perinatal development (Figure 1c, Appendix A). EGL width was significantly reduced to 32.46 ± 1.08 μm in the Term animals compared to earlier gestational time points (E-Pre and M-Pre), but there was no difference between E-Pre and M-Pre neonates (55.92 ± 3.71 μm in E-Pre, 54.32 ± 5.56 μm in M-Pre; *n* = six slices from three animals in each condition; *p* = 0.0006, one-way ANOVA, Tukey post hoc; Figure 1d). Similarly, cell density in the EGL was significantly decreased in the Term condition (*n* = 6 slices from 3 animals in each condition; *p* = 0.0028, one-way ANOVA, Tukey post hoc; Appendix A). In humans, drastic reduction in EGL width and cell density occurs after birth (through postnatal 4 months) [15], while EGL refinement occurred between M-Pre and Term in utero in the baboon cerebellum. In this period, IGL width was increased from 147.4 ± 7.07 μm in E-Pre to 222.3 ± 11.48 μm in M-Pre and 213.2 ± 25.02 μm in Term neonates (*n* = 6 slices from 3 animals in each condition; *p* = 0.0125, one-way ANOVA, Tukey post hoc; Appendix A). IGL cell density was constant throughout gestation in the E-Pre, M-Pre, and Term animals (*n* = 6 slices from 3 animals in each condition; *p* = 0.0877, one-way ANOVA, Tukey post hoc; Appendix A). Notably, these data indicate that granule cells actively migrate from the EGL to the IGL after the M-Pre time point during the last trimester, and the IGL progressively develops throughout the last trimester during normal gestational development.

Purkinje cells were identified by their expression of calbindin, a calcium-binding protein that is not expressed by other cells in the cerebellum (Figure 1c). The dendritic tree of Purkinje cells extends through the molecular layer (ML), and thus determines ML width. ML width was 40.71 ± 2.74 μm in E-Pre animals, and increased throughout development to ultimately become 162.4 ± 7.00 μm in Term animals (*n* = 6 slices from 3 animals from each condition; *p* < 0.0001, one-way ANOVA, Tukey post hoc; Figure 1e), similar to what has been reported previously [13]. The density of Purkinje cells did not significantly change throughout gestational development from EP to Term (*n* = 6 slices from 3 animals from each condition; *p* = 0.48, one-way ANOVA; Figure 1f). The soma diameter of Purkinje cells was increased to 24.73 ± 0.79 μm in Term animals compared to earlier gestational ages (from 20.41 ± 0.73 μm in E-Pre and 19.89 ± 0.57 μm in M-Pre; *n* = 18 cells from 3 E-Pre animals, 24 cells from 3 M-Pre animals, and 21 cells from 3 Term animals; *p* < 0.0001, one-way ANOVA, Tukey post hoc; Figure 1g). The results suggest that the soma and dendrites of Purkinje cells are developed during the last trimester without changes in Purkinje cell density. 

### 3.2. Preterm Birth Followed by NICU Experience Impacts Morphological Purkinje Cell Development

Next, we examined how premature birth and NICU experience impacts morphological features of Purkinje cell development. There were no significant effects of NICU experience on EGL and IGL development in the baboon cerebellum (Appendix A). While ML width was increased throughout the last trimester of pregnancy, it was thinner in NICU-experienced neonates compared to age-matched M-Pre controls (61.5 ± 2.61 μm, *n* = 6 slices from 3 NICU animals, vs. 91.9 ± 7.73 μm, *n* = 5 slices from 3 M-Pre animals, *p* < 0.01, Tukey post hoc; Figure 1e), corroborating previous studies [13,14]. This finding indicates alteration in dendritic development of Purkinje cells in the NICU-experienced group. However, NICU experience did not impact soma size (*n* = 18 cells from 4 NICU animals) or density of Purkinje cells (*n* = 6 slices from 3 NICU animals) compared to age-matched M-Pre controls (*p* > 0.05, one-way ANOVA, Tukey post hoc). Taken together, the results indicate that Purkinje cells undergo significant development in their morphological features including somatic size and dendrites in the last trimester of the pregnancy, and premature birth and NICU experience may impair or delay dendrite extension of Purkinje cells in the developing baboon cerebellum. 

### 3.3. NICU Experience Influences the Late-Gestation Refinement of Intrinsic Electrophysiological Properties of Purkinje Cells

Next, we examined how morphological development impacts physiological properties of Purkinje cells during perinatal development. In whole-cell patch clamp recordings, a prolonged depolarizing current injection (200 pA for 200 ms) resulted in action potentials (APs) in Purkinje cells from baboon neonates at different gestational ages (Figure 2a). Notably, Purkinje cell APs from term baboon neonates showed a strong adaption of AP firing whereas APs from E-Pre and M-Pre did not show a distinct adaption. The number of APs elicited by the same current injection (from 50 pA to 200 pA for 200 ms) was reduced in Purkinje cells from baboon neonates throughout perinatal development (*n* = 12 cells from 5 E-Pre animals, 8 cells from 4 M-Pre animals, and 3 cells from 3 Term animals; *p* < 0.0001, two-way ANOVA; Figure 2b). The NICU group showed a significant reduction of spikes compared with E-Pre (*n* = 9 cells from 6 NICU animals; *p* < 0.0001; Tukey post hoc), but no significant difference from M-Pre and Term Purkinje cells. The inter-spike interval (ISI), measured by the time difference between the peaks of the first and second APs, was not significantly changed during fetal development from E-Pre to Term neonates and in NICU-experienced neonates (*n* = 12 cells from 5 E-Pre animals, 7 cells from 4 M-Pre animals, 3 cells from 3 Term animals, and 9 cells from 6 NICU animals; *p* = 0.5514, One-way ANOVA; Figure 2c). Rheobase, the minimum amount of current necessary to induce an AP in Purkinje cells, increased throughout perinatal development, from 35.83 ± 4.68 pA in E-Pre animals to 120.0 ± 15.28 pA in Term animals (*n* = 12 cells from 5 E-Pre animals, 8 cells from 4 M-Pre animals, and 3 cells from 3 Term animals; *p* < 0.0001, one-way ANOVA; Figure 2d). In the NICU, Rheobase from Purkinje cells was between those from E-Pre and M-Pre, and was significantly smaller than those in M-Pre animals (*n* = 8 cells from 4 M-Pre animals and 9 cells from 6 NICU animals; *p* = 0.0272, Mann–Whitney test). 

Individual APs from different gestational ages were also analyzed (Figure 2a, insets). Threshold, amplitude, and half-width were measured using phase plots of each AP, the plot of dV/dt against membrane potential (Figure 2e). There was no significant change in AP threshold (*n* = 12 cells from 5 E-Pre animals, 8 cells from 4 M-Pre animals, and 3 cells from 3 Term animals; *p* = 0.1222, one-way ANOVA; Figure 2f) or amplitude throughout development (*n* = 12 cells from 5 E-Pre animals, 8 cells from 4 M-Pre animals, and 3 cells from 3 Term animals; *p* = 0.6488, one-way ANOVA; Figure 2g). There was, however, a decrease in AP half-width from 1.13 ± 0.08 ms in E-Pre animals to 0.44 ± 0.05 ms in Term animals (*n* = 12 cells from 5 E-Pre animals, 8 cells from 4 M-Pre animals, and 3 cells from 3 Term animals; *p* = 0.0081, one-way ANOVA; Figure 2h), indicating that aspects of Purkinje cell AP waveform are significantly modified during the third trimester of the pregnancy. In addition, we examined how NICU experience impacts AP properties of Purkinje cells. There was no significant effect on AP waveform in NICU animals in terms of threshold, amplitude, or half-width compared with age-matched M-Pre animals (*n* = 9 cells from 6 NICU animals). Taken together, Purkinje cells undergo fine-tuning of spiking properties throughout the last gestational period, and NICU experience may alter or delay some developmental features, including AP attenuation and rheobase, compared to development in the womb.

### 3.4. Synaptic Input to Purkinje Cells is Increased throughout Development and Impeded by NICU Experience

During gestational development of the cerebellum, a key event is migration of granule cells from EGL to IGL, during which time they form synapses onto Purkinje cells [7]. Morphological and physiological development of Purkinje cells may be regulated by synaptic input from granule cells [9]. Therefore, we examined how synaptic inputs to Purkinje cells are changed throughout the later gestational development by recording spontaneous postsynaptic currents (sPSCs; Figure 3a). sPSCs in E-Pre animals had amplitudes of 24.68 ± 2.64 pA and were completely blocked by CNQX, an AMPA receptor antagonist, indicating that these synaptic currents were AMPA receptor-mediated excitatory synaptic currents. sPSC amplitude was not significantly changed throughout development (*n* = 14 cells from 5 E-Pre animals, 9 cells from 4 M-Pre animals, and 3 cells from 2 Term animals; *p* = 0.9706, one-way ANOVA; Figure 3b). However, sPSC frequency increased throughout development, from 1.04 ± 0.23 Hz at E-Pre to 6.08 ± 2.063 Hz in Term animals (*n* = 14 cells from 5 E-Pre animals and 3 cells from 2 Term animals; *p* = 0.0002, one-way ANOVA, Tukey post hoc; Figure 3c), indicating that there is an increase in synaptic inputs in the later gestational period. sPSC frequency increased to 2.54 ± 0.26 Hz in M-Pre (*n* = 9 cells from 4 M-Pre animals), but did not increase in NICU (0.99 ± 0.26 Hz; *n* = 7 cells from 5 NICU animals). Thus, there was significant reduction in sPSC frequency in NICU compared with age-matched M-Pre (*p* = 0.0443, Mann–Whitney test). This suggests that synaptic inputs to Purkinje cells are not properly developed during NICU experience following premature birth, which may influence the structural and functional development of Purkinje cells. 

### 3.5. DCN Neurons of NICU-Experienced Animals have Altered AP Firing Properties

Purkinje cells are the sole output of the cerebellar cortex, integrating all signals within the cerebellar cortex and relaying the information to the DCN. Alterations in spiking properties and synaptic inputs to Purkinje cells in NICU-experienced neonates can therefore impact DCN neuronal activity [10,11]. To examine how pre-term birth followed by NICU experience impacts downstream signaling in the cerebellar circuitry, we recorded cells of the DCN using whole-cell patch clamp electrophysiology. DCN neurons in E-Pre, NICU, and M-Pre baboon neonates displayed spontaneous AP firing (Figure 4a). While frequency of spontaneous AP firing was not changed between E-Pre and M-Pre neonates, it was significantly increased in NICU (*n* = 4 cells from 3 E-Pre animals, 6 cells from 3 NICU animals, and 4 cells from 3 M-Pre animals; *p* = 0.0426, one-way ANOVA, Tukey post hoc; Figure 4b). The frequency of spontaneous spikes in DCN neurons is affected by inhibitory GABAergic input from Purkinje cells [12]. Thus, this hyperexcitability could be due to reduced inhibitory input from Purkinje cells. Additionally, intrinsic AP firing properties of DCN cells were assessed when the cells were depolarized via a current injection of 200 pA for 200 ms, resulting in a train of APs (Figure 4C). The number of APs elicited by the depolarization did not change between E-Pre, M-Pre, and NICU animals (*n* = 4 cells from 2 E-Pre animals, 7 cells from 4 NICU animals, and 4 cells from 3 M-Pre animals; *p* = 0.7825, one-way ANOVA; Figure 4d). There was no change in ISI during development or NICU experience (*n* = 4 cells from 2 E-Pre animals, 7 cells from 4 NICU animals, and 4 cells from 3 M-Pre animals; *p* = 0.4762, one-way ANOVA; Figure 4e), and rheobase was also not changed in the E-Pre, M-Pre, and NICU conditions (*n* = 4 cells from 2 E-Pre animals, 7 cells from 4 NICU animals, and 4 cells from 3 M-Pre animals; *p* = 0.6197, one-way ANOVA; Figure 4f). Individual APs from DCN cells were also analyzed (Figure 4c, insets). There was no significant change in AP threshold between E-Pre, M-Pre, and NICU animals (*n* = 4 cells from 2 E-Pre animals, 7 cells from 4 NICU animals, and 4 cells from 3 M-Pre animals; *p* = 0.9593, one-way ANOVA; Figure 4g). AP amplitude did not significantly change between E-Pre and M-Pre, but was significantly reduced in NICU compared to age-matched controls (*n* = 4 cells from 2 E-Pre animals, 7 cells from 4 NICU animals; *p* = 0.0301, one-way ANOVA, Tukey post hoc; Figure 4h). Similarly, there was no significant difference in AP halfwidth between E-Pre and M-Pre DCN cells, but there was an increase in AP halfwidth in NICU DCN cells compared to M-Pre cells (*n* = 4 cells from 2 E-Pre animals, 7 cells from 4 NICU animals, and 4 cells from 3 M-Pre animals; *p* = 0.0129, one-way ANOVA, Tukey post hoc; Figure 4i). This change in AP halfwidth is not likely to contribute to the increase in spontaneous AP firing frequency increase seen in NICU DCN cells, as increased AP halfwidth is typically associated with a decrease in the ability of cells to fire high frequency APs. Thus, the hyperexcitability of DCN cells seen in the NICU condition as measured by spontaneous AP firing is not due to changes in intrinsic properties, but rather that Purkinje cell output to the DCN may be the primary factor influencing DCN hyperexcitability.

## 4. Discussion

Although preterm birth is still common worldwide, its effects on the cellular physiology of the central nervous system have been understudied. Cerebellar Purkinje cells are specifically susceptible to developmental disruptions, as the cerebellum is a late-developing brain structure that is critically impacted by preterm birth. The present study demonstrates that Purkinje cell structure and function are drastically developed during the last trimester of the pregnancy and that development in the NICU rather than in utero impedes developmental features in a baboon model of preterm birth. Alterations in Purkinje cell dendrite extension and action potential waveform in the NICU-experienced neonates are associated with reduced excitatory synaptic input from granule cells and DCN hyperexcitability.

The findings in the baboon cerebellum at different gestational ages compare to postnatal development of Purkinje cells in rodent models. The morphology of rat Purkinje cells at postnatal day 6 (P6)-P9 and P12 resemble baboon Purkinje cells at the E-Pre and M-Pre stages, respectively, with increasing extension and ramification of processes through P30 [16], similar to the third trimester of the gestational development of baboon Purkinje cells through full term birth. In baboon cerebellum, the most prominent morphological feature of Purkinje cell development during the third trimester of gestation was an increase in dendritic size and complexity, resulting in an increase in ML width (Figure 1). The developmental course of morphological features is comparable to P6-P30 in rat cerebellum; however, comparison of electrophysiological properties of Purkinje cells are more variable. While the threshold and amplitude of action potentials remained stable, rheobase significantly increased throughout late gestational development in baboon Purkinje cells, similar to the observed Purkinje cell rheobase in P9-P30 rats [16]. In rat Purkinje cells, one signature physiological feature during early postnatal development is an emergence of repetitive bursts of spikes and an increase in spike frequency [16]. In baboon Purkinje cells, action potential half-width was drastically reduced and the inter-spike-interval showed a trend of reduction throughout late gestation, indicating that Purkinje cells undergo drastic changes in morphology and physiological properties during the last trimester in utero. We would like to state that there were limitations in obtaining a number of intact and acute baboon brain slices at different gestational time points for electrophysiological recordings. Although ISI and threshold showed a developmental trend, there was no significant difference. Thus, it is worthy to examine physiological properties of Purkinje cells at more variable GA from M-Pre and Term neonates.

The comparison of Purkinje cell development between species highlights the importance of studying brain development in primates, which is similar to that of humans in that it occurs primarily during gestation, as opposed to postnatally, as in rodents. For example, the human Purkinje dendritic arbor continues to develop its characteristic shape after birth [19,20,21], which corroborates the marked arborization seen at term birth compared to earlier time points in the baboon. The first synapses on somatic spines of Purkinje cells become prominent at 18 to 24 weeks of gestation in humans, before the arborization of the Purkinje dendrites [20]. Similar to what was observed in the baboon, Purkinje arborization was associated with an increase in ML width in late gestation and early postnatal stages in humans [15]. A histological analysis of postmortem human neonate tissue after preterm birth showed that EGL cell density and thickness were altered by preterm birth and ex-utero effects whereas ML thickness are not affected [15]. In baboon neonates, we found that preterm birth and NICU experience impact ML thickness without significant changes in EGL width and cell density. This contrast could be due to the time for collecting tissues. Cerebellar tissues from human studies are variable in their gestational or postnatal time points. In the human cerebellum, ML thickness was stable during late gestation and dramatically increased after term birth (>39 weeks GA) [15], whereas the baboon cerebellum displayed a significant increase in ML thickness from E-Pre to M-Pre and Term neonates [13] (Figure 1). Therefore, preterm birth and NICU experience may more strongly influence Purkinje dendritic development and ML thickness in baboon cerebellum than in human. In the human cerebellum, preterm birth and ex-utero effects impact EGL and IGL thickness and cell density indicating altered migration of granule cells from the EGL to IGL [15]. In baboon cerebellar development, EGL thickness and cell density were significantly changed in M-Pre neonates (delivered at 80% of GA) and Term neonates. Thus, it is expected that NICU experience between M-Pre and Term may critically impact EGL thickness and cell density in baboon. The effects of preterm birth on layer widths and cerebellar size have been documented previously in both humans and baboons [13,14,15]. It is important to investigate the impacts of those structural changes on the functional cellular output of the cerebellar cortex.

The present study defines the increase in synaptic input from granule cells to Purkinje cells throughout development, as well as the impedance of this synaptic input following extreme preterm birth and NICU experience. Climbing fiber-Purkinje cell synapse pruning occurs during the late gestational developmental stages assessed in this study, and deficits in pruning of this synapse have been linked to tremor and neurological degenerative disorders [22,23]. The effect of preterm birth on climbing fiber pruning has, to our knowledge, not been examined and would be a worthwhile study. The relationship between Purkinje cells and granule cells migrating from the EGL to IGL are crucial for their reciprocal development. Granule cell neurons are critical for the development and survival of Purkinje cells [24,25]. Purkinje cells release Sonic hedgehog, which signals through Gli1 and Gli2 on EGL cells to influence granule cell development, as neutralization of Sonic hedgehog results in a reduced EGL cell proliferation [21,26]. Therefore, premature birth and NICU experience could influence the reciprocal development of EGL, IGL, and ML, resulting in the well-documented cerebellar volume reduction due to preterm birth [2].

Mechanisms of reduced cerebellar size due to preterm birth generally include subarachnoid hemorrhage or hypoxia. Cerebellar volume has shown to be reduced in a neonatal mouse model of cerebellar hemorrhage [27]. In infants born preterm, blood over the cerebellum due to subarachnoid hemorrhage, accompanied by hemosiderin deposition, has been shown to result in reduced cerebellar volume [28,29] and potentially direct effects on Purkinje cells in preterm newborns, as discussed by Volpe [7]. Hypoxemia in fetal sheep resulted in a reduced number of Purkinje cells [30,31], and chronic hypoxia in neonatal mice resulted in altered Purkinje cell structure and functional locomotor deficits [32,33]. In the present study, preterm baboon neonates were born via cesarean section without any perinatal complications. We must take careful consideration to compare baboon neonate tissue and human postmortem tissue, which may have the possibility of pre-existing complications and potential abnormalities beyond premature birth. Additionally, there may be influences on cerebellar structure and function that are impacted by environmental aspects of NICU experience rather than preterm birth itself [34]. In this study, NICU-experienced baboon neonates were treated with all medications, hormones, and nutrients, which were administered in the NICU to support viability, similar to what is administered to preterm human infants. The effects of these interventions on cerebellar development are yet to be determined and could confound the results of this study. Any possible effects of extreme preterm birth independent from NICU experience were not assessed in this study, as NICU interventions were necessary for the survival of the neonates.

Preterm birth in humans has been shown severely disrupt cerebellar development, associated with long-term deficits in cognition and motor function [5], as well as neuropsychological well-being [8]. However, the cellular functional deficits in the cerebellum due to preterm birth are poorly understood. The present study using non-human primate baboon neonates, with controlled conditions and environments, are critical for understanding neurodevelopment at discrete time points during gestation and the effects of preterm birth. We demonstrate functional changes in critical cellular players in the cerebellar circuit in the preterm baboon, including reduced excitatory input to Purkinje cells from granule cells accompanying deficits in Purkinje cell physiology and DCN cell output. This work provides insight into the cellular physiology that may contribute to cerebellar-related deficits resulting from preterm birth.

## 5. Conclusions

Baboon cerebellum undergoes developmental refinement during late gestation, and NICU experience following extreme preterm birth impacts cellular development in the cerebellum that can lead to functional deficits. 

## Figures and Tables

**Figure 1 brainsci-10-00897-f001:**
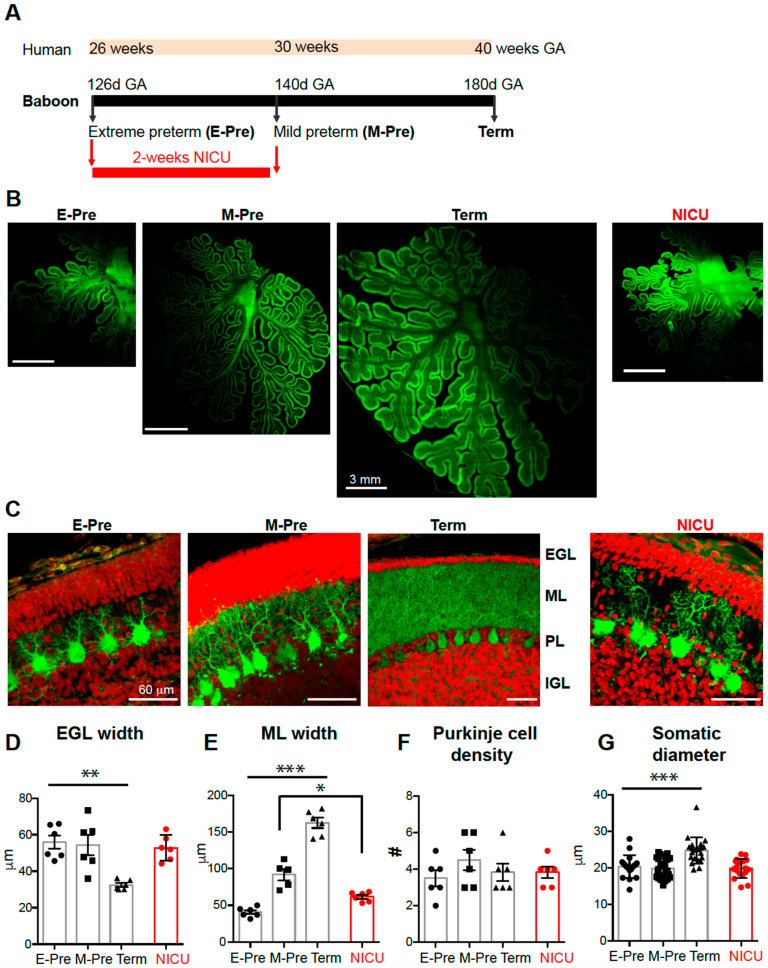
Morphological development of baboon cerebellum during the last trimester of the pregnancy. (**A**) Experimental paradigm comparing baboon gestational age (GA) compared to human GA. (**B**) Cerebellar sections from E-Pre, M-Pre, Term, and NICU animals immunostained to show Purkinje cell bodies, dendrites, and axons expressing calbindin (CB, green). (**C**) Granule cell nuclei stained with DAPI (red) in the external granule layer (EGL) and internal granule layer (IGL), with CB labeling of Purkinje dendrites in the molecular layer (ML) and Purkinje cell bodies in the Purkinje layer (PL) in E-Pre, M-Pre, Term, and NICU cerebellar sections. (**D**–**G**) Summary of EGL width (**D**), ML width (**E**), Purkinje cell density (**F**), and Purkinje cell soma diameter (**G**) at each gestational age. Data are represented as ± SEM. *, **, and *** represent *p* < 0.05, *p* < 0.01, and *p* < 0.001, respectively.

**Figure 2 brainsci-10-00897-f002:**
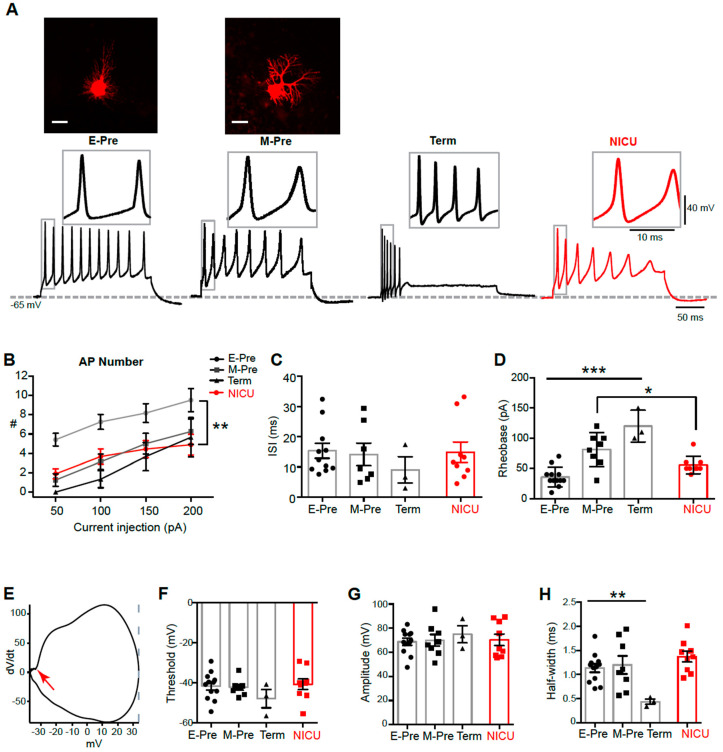
Purkinje cell intrinsic properties are altered in the NICU condition. (**A**) Top, confocal images of Purkinje cells Alexa dye-filled during whole-cell recording. Scale = 20 m. Bottom, representative traces of action potentials (APs) induced by 200 pA current injections from Purkinje cells from E-Pre, M-Pre, Term, and NICU baboon neonates. Inset, expanded view of action potentials in box. (**B**–**D**) Summary of AP number (**B**) in response to current injection from 50 to 200 pA, inter-spike interval (ISI; **C**), and rheobase (**D**) from each group. (**E**) Representative phase plot of an AP (dV/dt), demonstrating the threshold (red arrow), peak (grey line), and amplitude (voltage difference from the threshold to the peak). (**F**–**H**) Summary of threshold (**F**), amplitude (**G**), and halfwidth (**H**) of a single AP from each gestational age. Data are represented as ± SEM. *, **, and *** represent *p* < 0.05, *p* < 0.01, and *p* < 0.001, respectively.

**Figure 3 brainsci-10-00897-f003:**
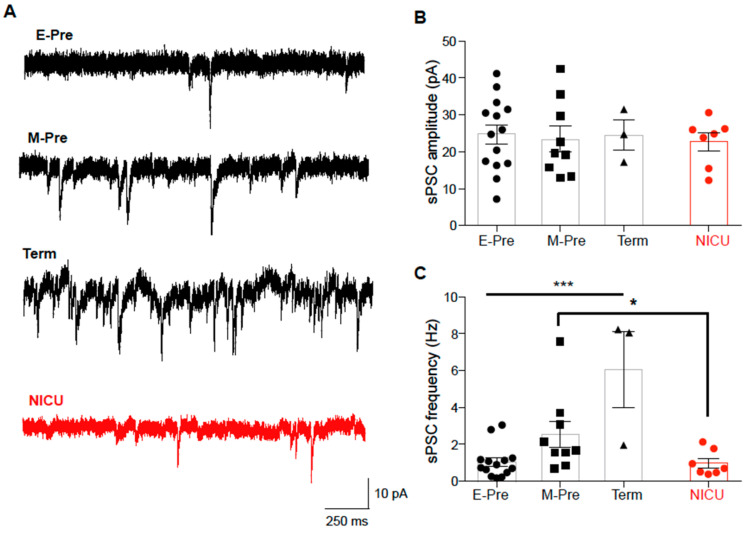
Spontaneous postsynaptic currents (sPSCs) increase in frequency throughout gestation and are reduced after NICU experience. (**A**) Representative traces of sPSCs recorded from Purkinje cells of E-Pre, M-Pre, Term, and NICU neonates. (**B, C**) Summary of sPSC amplitude (**B**) and frequency (**C**) at each group. Data are represented as ± SEM. * and *** represent *p* < 0.05, and *p* < 0.001, respectively.

**Figure 4 brainsci-10-00897-f004:**
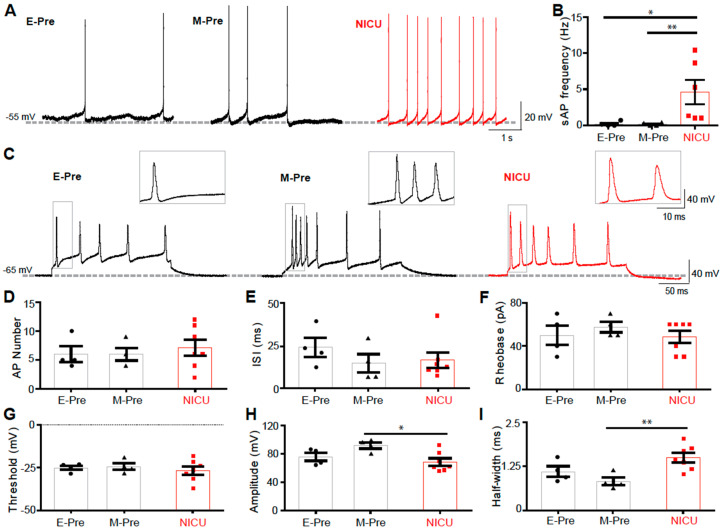
DCN cell APs are impacted by NICU experience. (**A**) Spontaneous AP recordings from DCN neurons from E-Pre, M-Pre, and NICU baboon neonates. (**B**) Summary of spontaneous AP frequency in DCN neurons from each group. (**C**) Representative traces of action potentials induced by 200 pA current injections from DCN neurons from E-Pre, M-Pre, and NICU animals. (**D**–**I**) Summary of DCN neuron AP number (**D**), inter-spike interval (ISI; **E**), rheobase (**F**), threshold (**G**), amplitude (**H**), and half-width (**I**) from each group. Data are represented as ± SEM. *, and ** represent *p* < 0.05, and *p* < 0.01, respectively.

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
