# Peer review of "Preterm Birth Impedes Structural and Functional Development of Cerebellar Purkinje Cells in the Developing Baboon Cerebellum"

_brainsci, 2020, doi:10.3390/brainsci10120897_

Round 1

Reviewer 1 Report

In a non-human primate baboon model, the authors tried to investigate the effects of preterm birth and NICU experience on cerebellar development.

The authors could show gestational age dependent thickness of the EGL, the IGL and the ML of the cerebellar cortex. Electrophysiological analysis reveals gestational age dependent functionality of cerebellar Purkinje cells. ML development has been shown previously.

Here are my questions:

Which lobules did the authors use for morphological analyses? Did the authors use the same lobules in each sample? Neuronal development in the cerebellum is lobule dependent.

How did the authors distinguish between effects of extreme preterm birth and NICU on cerebellar development? There was no control group without the effects of intensive care unit.

Extreme preterm birth might have affected Purkinje cell development even without NICU exposure.

How could the authors verify, if NICU impairs or improves cerebellar development after extreme preterm birth?

The number of samples for electrophysiological analysis seems to be very low. Three cells out of two animals is not appropriate for interpretation.

The author postulate preterm birth might affect cerebellar volume. Several studies showed decreased cerebellar volume after preterm birth.

The authors could not show the impact of neonatal intensive care unit experience on cerebellar development. The authors showed the effects of extreme preterm birth followed by NICU exposure on cerebellar development.

Author Response

Reviewer#1

Which lobules did the authors use for morphological analyses? Did the authors use the same lobules in each sample? Neuronal development in the cerebellum is lobule dependent.

Response) Morphological analysis was quantified in lobule 8. We have added this information to the Methods section.

How did the authors distinguish between effects of extreme preterm birth and NICU on cerebellar development? There was no control group without the effects of intensive care unit. Extreme preterm birth might have affected Purkinje cell development even without NICU exposure. How could the authors verify, if NICU impairs or improves cerebellar development after extreme preterm birth?

Response) In this study, we compare cerebellar development age-matched baboon neonates in the womb vs in the NICU. The birth of these neonates via cesarean section followed the same protocol at extreme preterm and mild preterm time points. Extreme preterm-born baboon neonates could not survive without NICU intervention, and thus it was not possible to compare NICU experience to development outside the NICU after preterm birth.

We address this caveat to the Discussion section: “…there may be influences on cerebellar structure and function that are impacted by environmental aspects of NICU experience rather than preterm birth itself (Noguchi et al., 2008). In this study, NICU-experienced baboon neonates were treated with all medications, hormones, and nutrients, which were administered in the NICU to support viability, similar to what is administered to preterm human infants. The effects of these interventions on cerebellar development are yet to be determined and could confound the results of this study. Any possible effects of extreme preterm birth independent from NICU experience were not assessed in this study, as NICU interventions were necessary for the survival of the neonates”.

The number of samples for electrophysiological analysis seems to be very low. Three cells out of two animals is not appropriate for interpretation.

Response) Due to limitations (e.g. animal cost) in obtaining acute non-human primate brain tissues for electrophysiological recordings, we cannot increase the number of samples at this time.

The author postulate preterm birth might affect cerebellar volume. Several studies showed decreased cerebellar volume after preterm birth.

Response) We have edited the statement to more clearly state that the preterm birth on cerebellar volume are well-documented.

The authors could not show the impact of neonatal intensive care unit experience on cerebellar development. The authors showed the effects of extreme preterm birth followed by NICU exposure on cerebellar development.

Response) We agree with the reviewer and have edited the manuscript to address this point in the Discussion section, as mentioned above.

Reviewer 2 Report

This is an important and well-designed study that identifies significant structural (predominantly ML thickness) and functional (GC-PC synaptic connectivity and DCN excitability) in cerebella from ventilated pre-term baboons.  This finding suggests that cerebellar development is highly sensitive process during late gestation in the baboon, which has implications for human. 

The main critique is on completeness:  Since climbing fiber pruning is another key and sensitive developmental stage that may lead to neurological consequences it would be a pity not to have analyzed these features in this study.  

One minor critique is the lack of detail of the acquisition of the sagittal sections for morphometry.  Since orientation is key for measuring height of ML the site of the cerebellum (mid sagittal or not) and steps taken to orient the sagittal sections should be better clarified.  

Author Response

The main critique is on completeness:  Since climbing fiber pruning is another key and sensitive developmental stage that may lead to neurological consequences it would be a pity not to have analyzed these features in this study.  

Response) We focused on Purkinje cell development. Although climbing fiber pruning is an important feature during cerebellum development, it is out of scope in this manuscript. However, we have addressed the possible effects of preterm birth on climbing fiber pruning and consequential neurological deficits in the Discussion section: “Climbing fiber-Purkinje cell synapse pruning occurs during the late gestational developmental stages assessed in this study, and deficits in pruning of this synapse have been linked to tremor and neurological degenerative disorders (Kuo et al., 2017; Pan et al., 2020). The effect of preterm birth on climbing fiber pruning has, to our knowledge, not been examined and would be a worthwhile study.”.

One minor critique is the lack of detail of the acquisition of the sagittal sections for morphometry.  Since orientation is key for measuring height of ML the site of the cerebellum (mid sagittal or not) and steps taken to orient the sagittal sections should be better clarified.  

Response) All cerebellar sections were midsagittal, and this has been clarified in the manuscript.

Round 2

Reviewer 1 Report

The authors edited the statement that preterm birth can effect cerebellar volume. The authors should confirm this statement by an appropriate reference.

For example:

Limperopoulos C, Soul JS, Gauvreau K, Huppi PS, Warfield SK, Bassan H,

Robertson RL, Volpe JJ, du Plessis AJ. Late gestation cerebellar growth is rapid

and impeded by premature birth. Pediatrics. 2005 Mar;115(3):688-95. doi:

10.1542/peds.2004-1169. PMID: 15741373

Author Response

We added the following reference.

Limperopoulos C, Soul JS, Gauvreau K, Huppi PS, Warfield SK, Bassan H,

Robertson RL, Volpe JJ, du Plessis AJ. Late gestation cerebellar growth is rapid

and impeded by premature birth. Pediatrics. 2005 Mar;115(3):688-95. doi:

10.1542/peds.2004-1169. PMID: 15741373